# Violent Video Recognition by Using Sequential Image Collage

**DOI:** 10.3390/s24061844

**Published:** 2024-03-13

**Authors:** Yueh-Shen Tu, Yu-Shian Shen, Yuk Yii Chan, Lei Wang, Jenhui Chen

**Affiliations:** 1Department of Electrical Engineering, Chang Gung University, Taoyuan 33302, Taiwan; robtu328@gmail.com; 2Department of Computer Science and Information Engineering, Chang Gung University, Taoyuan 33302, Taiwan; ivan880310@gmail.com (Y.-S.S.); chanyukyii@hotmail.com (Y.Y.C.); 3School of Software, Dalian University of Technology, Dalian 116024, China; lei.wang@dlut.edu.cn; 4Division of Breast Surgery and General Surgery, Department of Surgery, Chang Gung Memorial Hospital, Linkou 33375, Taiwan; 5Department of Electronic Engineering, Ming Chi University of Technology, Taishan District, New Taipei City 24301, Taiwan

**Keywords:** training, image recognition, neurons, computer architecture, multilayer perceptrons, Transformers, behavioral sciences

## Abstract

Identifying violent activities is important for ensuring the safety of society. Although the Transformer model contributes significantly to the field of behavior recognition, it often requires a substantial volume of data to perform well. Since existing datasets on violent behavior are currently lacking, it will be a challenge for Transformers to identify violent behavior with insufficient datasets. Additionally, Transformers are known to be computationally heavy and can sometimes overlook temporal features. To overcome these issues, an architecture named MLP-Mixer can be used to achieve comparable results with a smaller dataset. In this research, a special type of dataset to be fed into the MLP-Mixer called a sequential image collage (SIC) is proposed. This dataset is created by aggregating frames of video clips into image collages sequentially for the model to better understand the temporal features of violent behavior in videos. Three different public datasets, namely, the dataset of National Hockey League hockey fights, the dataset of smart-city CCTV violence detection, and the dataset of real-life violence situations were used to train the model. The results of the experiments proved that the model trained using the proposed SIC is capable of achieving high performance in violent behavior recognition with fewer parameters and FLOPs needed compared to other state-of-the-art models.

## 1. Introduction

With the advancement in artificial intelligence technology, recent developments in deep learning have improved significantly in the accuracy and efficiency of human action recognition (HAR) in closed-circuit television (CCTV) [1,2]. Among the many deep learning architectures developed, convolutional neural networks (CNNs) [3,4] are particularly effective at detecting and recognizing patterns in video data [5], making them well-suited for this task. Recurrent neural networks (RNNs) [6] are also used to capture time-related relationships between video frames. Despite the help of CNNs and RNNs, the performance of deep learning models for behavior recognition is still hindered by several challenges. One of the key challenges in HAR involves the large variability in human actions [7], including different viewpoints [8,9], clothing [10,11], lighting conditions [12], and occlusions [13]. To overcome this challenge, researchers have developed various techniques, such as data augmentation [14], transfer learning [15], and ensemble methods [16], to improve the robustness and generalizability of HAR models.

Detecting violent actions within human activity recognition (HAR) is of paramount importance, serving as a crucial tool to alert security officials and law enforcement to potential threats, thereby safeguarding the community. Identifying violent behavior within video data from surveillance cameras—encompassing actions like physical altercations, aggressive behaviors, and vandalism—is a formidable challenge in HAR [17,18,19]. To tackle this challenge, researchers have innovated various techniques. These range from motion-based methodologies [20] and appearance-centric approaches [21] to spatio-temporal strategies, harnessing nuances like motion patterns [22], color histograms [23], and the interplay between spatial and temporal elements within the scene [24,25].

A fundamental requirement for the efficacy of these methods is an ample corpus of labeled data for training. However, the dearth of labeled violent video segments (ViVis) hampers the training of robust violence recognition models tailored for CCTV systems [26]. This paucity impedes the development of models adept at recognizing infrequent or emerging actions that are violent. To overcome this challenge, various techniques have been explored by researchers, such as unsupervised learning, which involves the use of tools like generative adversarial networks (GANs) and clustering [27,28] to derive representations that are meaningful from videos that do not have explicit labels. An alternative and promising avenue is transfer learning [29], wherein a model pretrained on a broader dataset [30] is adjusted and refined using a violent dataset that is labeled and reduced in size, enhancing both accuracy and adaptability in scenarios with restricted availability of labeled data.

While the architecture of vision Transformer (ViT) [31] has garnered acclaim across various visual tasks, its adoption in violence recognition is not without challenges. Notably, ViT’s expansive model size can hinder the deployment on constrained devices or in real-time applications [32]. Furthermore, its frame-by-frame feature extraction might compromise temporal context [33]. To mitigate these concerns, innovations like the multilayer perceptron mixer (MLP-Mixer) [34] have emerged. These streamlined models promise expedited inference and reduced computational overhead, making them viable for deployment on devices like embedded systems, smartphones, and energy-efficient cameras. Additionally, the efficacy of the MLP-Mixer in capturing temporal dynamics within HAR has been empirically validated [35].

Fan et al. introduced an innovative approach, termed the super image for action recognition (SIFAR) [36], aimed at enhancing temporal feature extraction. SIFAR transforms input video frames into composite images, streamlining action recognition akin to image classification. While this amalgamation of sequential frames into a singular, comprehensive image captures temporal nuances, the subsequent resizing sacrifices detailed information fidelity. To counteract this drawback, this study reintroduces the primary video frames, constructing a hybrid dataset to restore this eroded granularity.

In pursuit of computational efficiency, a backbone network centered around the MLP-Mixer architecture for video violence recognition is advocated in this paper. Confronting the limited scope of the video violence dataset, a novel data augmentation technique—sequential image collage (SIC)—is presented in this research. The SIC amalgamates the super images from both SIFAR [36] and original video frames, circumventing the need for the intricate feature design typical of CNNs or the data-intensive nature associated with Transformers.

Specifically, the primary contributions of this paper are listed as follows.

Proposing an MLP-Mixer-driven framework for violence recognition, distinguished by its reduced computational demands vis-à-vis Transformer-centric models.Introducing a composite dataset comprising both image collages that capture the space and time relationships between video frames and unmodified video frames. This composite dataset augments the training process, culminating in a spatio-temporal model exhibiting superior action recognition capabilities.

The rest of this paper is structured in the following manner: Section 2 describes the development of HAR from handcrafted features and deep learning to Transformers. Section 3 illustrates the SIC architecture proposed in this work. Section 4 demonstrates the efficiency of SIC, with Section 5 finally concluding the paper.

## 2. Related Work

### 2.1. Handcrafted Feature-Based Methods

In the nascent stages of computer vision research, prior to the deep learning revolution, a predominant focus was placed on devising handcrafted features. The rationale behind this approach was grounded in the quest to discern specific visual motifs within videos that served as telltale signs of the depicted action or event. Once these discernible patterns were identified, they were meticulously encoded into a structured set of features, primed for consumption by classification models tasked with predicting the observed actions or events.

Among the many handcrafted featured methodologies, the dense trajectories (DTs) method [37] stands out as particularly influential. This seminal technique initiates its workflow by computing dense optical flow fields for every frame within a given video sequence. Subsequently, it embarks on a trajectory-tracking endeavor, meticulously following the paths of salient points across a series of frames. Along these trajectories, a triad of essential feature descriptors is computed: the motion boundary histograms (MBHs), the histogram of optical flow (HOF) [38], and the histogram of oriented gradients (HOG) [39]. Collectively, these descriptors encapsulate both the local appearance nuances and intricate motion dynamics inherent to the video content.

To distill these rich feature sets into a more manageable and uniform representation, the bag of features (BoF) encoding strategy is employed. This culminates in the derivation of a fixed-length feature vector, which is uniquely representative of each video. As a denouement, these compacted feature vectors are channeled into a support vector machine (SVM) [40] for the overarching training process.

Building upon the foundational principles of the DT methodology, the enhanced dense trajectories (iDTs) method [23] emerges as a refined iteration. iDT incorporates the speeded up robust features (SURF) [41]—a computationally expedited variant of the scale-invariant feature Transform (SIFT) [42]—to fortify its feature extraction prowess. Moreover, in lieu of the traditional BoF encoding, iDT adopts the more sophisticated fisher vector (FV) encoding technique, further elevating the discriminative power and robustness of the feature representations.

### 2.2. Deep Learning-Based Methods

Historically, techniques using handcrafted features have dominated the scene. However, the advent of deep learning ushered in a paradigm shift towards the autonomous extraction of features from raw video data. Central to this evolution was the emergence of the two-stream convolutional neural network (CNN), which quickly gained prominence. Simonyan et al. pioneered this arena, introducing a dual-stream CNN method [43] that concurrently amalgamated spatial and temporal data, with the streams representing RGB flow and optical flow, respectively. Building on this foundation, Feichtenhofer et al. [44] devised an innovative fusion technique, enhancing the integration of intricate spatio-temporal nuances across two-stream CNNs. Meanwhile, Ng et al. [45] integrated CNN and LSTM architectures, leveraging the CNN for feature extraction and subsequently channeling these features to the LSTM [46] for further analysis. Wang et al. [47] advanced the field with the temporal segment network (TSN), segmenting videos into distinct parts, extracting frames from each segment, and culminating in a composite prediction.

Meanwhile, the realm of violent behavior recognition in videos has witnessed escalating interest. To discern violent behavior in videos, Bermejo et al. [17] melded local and global spatio-temporal features and introduced the hockey fight dataset—a compilation of 1000 NHL ice hockey action clips. Hassner et al. [18] charted a fresh course, evaluating temporal variations in flow-vector magnitudes. They encapsulated these dynamics using violent flow (ViF) descriptors, subsequently employing a linear support vector machine (SVM) to decide whether the data are violent or not by performing binary classification. Further enriching this domain, Xu et al. [20] devised a potent framework grounded in the MoSIFT algorithm and a sparse coding strategy, demonstrating efficacy across both dense and sparse scenes within violent behavior datasets.

### 2.3. Transformer-Based Methods

In recent times, Transformers [48] have garnered significant attention, consistently delivering superior outcomes across diverse domains. This surge in Transformer-based advancements has spurred researchers to tailor architectures to specific requirements. For instance, Dosovitskiy et al. [31] introduced the vision Transformer (ViT), a groundbreaking image recognition model. ViT dissects input images into manageable patches, subsequently treating these patches as token sequences for Transformer processing. This approach has set new benchmarks on various standard datasets.

Building upon the ViT framework, the video vision Transformer (ViViT) was unveiled by Arnab et al. [49]. This innovation disassembles videos into both spatial and temporal patches, channeling them through a ViT-centric backbone network. Notably, ViViT integrates an avant-garde temporal attention mechanism adept at capturing interframe dynamics. Such enhancements have propelled ViViT to the forefront of video recognition, showcasing exemplary results on renowned datasets like kinetics-400 [50] and something-something V1 and V2 [51].

In a parallel development, Liu et al. [52,53] introduced the swin Transformer, originally tailored for image recognition. This model swiftly established itself as a benchmark, outperforming contemporaneous architectures. Expanding its versatility, the video swin Transformer was devised, enabling the processing of spatio-temporal video patches as token sequences. Mirroring its predecessor’s success, the video swin Transformer has also clinched top-tier performance across multiple video recognition benchmarks.

Aside from modifying the Transformer itself, previous studies also tried to improve the model’s performance by altering the input’s architecture. An example can be seen with the proposal of a module called the temporal shift module (TSM) by Lin et al. [54] to improve the efficacy of video understanding. The TSM increases the diversity and content of video data by shifting the video data on the time axis. Combined with a Transformer, this method greatly improves the speed and efficiency of video understanding.

Despite the advancements in Transformers, they can be inefficient and computationally heavy when the dataset provided for training is too large [32]. For this reason, the MLP-Mixer is proposed to counter these challenges [34]. However, having single images as input in the MLP-Mixer can still result in information loss in temporal features as each video frame is related to the previous and next consecutive frame. Hence, inspired by Fan et al.’s work on the use of super images [36], a special input called the sequential image collage is introduced in the next section where the frames would be arranged in a 3 × 3 collage to better capture the temporal features of the input video frames.

## 3. Methodology

The proposed SIC is illustrated in detail in this section. The architecture of the SIC is shown in Figure 1.

### 3.1. Main Architecture

In this paper, the initial step involves extracting individual frames from the video sequence. Subsequently, these frames are amalgamated into composite images, effectively creating an image collage that encapsulates both spatial and temporal dimensions. This approach ensures that each collage encapsulates a continuum of information over a specific time frame. These synthesized images, in conjunction with the original frames to account for any potential information loss, are then fed into the MLP-Mixer backbone architecture. Ultimately, based on these processed data, the model determines whether the given collage depicts violent or non-violent content. A visual representation of the comprehensive architecture can be observed in Figure 1.

### 3.2. Dual-form Dataset

In prior research, training typically relied on individual images, overlooking the interconnectedness between consecutive video frames. Given that adjacent frames in a video sequence often possess temporal correlations, neglecting this inherent relationship could compromise training efficacy. To address this, we adopt the approach delineated in [36]. Initially, videos are decomposed into individual frame images. Subsequently, sets of nine consecutive frames are amalgamated to form a unified 3×3 image collage, as illustrated in Figure 2.

### 3.3. MLP-Mixer

While both the mechanism of self-attention in Transformers and the operation of convolutions in CNNs have demonstrated their efficacy in numerous studies, they are not indispensable. The MLP-Mixer emerges as a compelling alternative, leveraging only MLPs yet delivering commendable performance across various image recognition tasks due to its inherent simplicity.

The foundational step mirrors that of the vision Transformer: each input is fragmented into contiguous and disjoint patches, initiating the patch embedding process. These embedded features subsequently traverse through a Mixer Layer. This layer houses two distinct MLP categories: channel-mixing MLPs and token-mixing MLPs, with each serving a unique purpose in data fusion.

Token-mixing MLPs are adept at deciphering spatial intricacies within the image, facilitating the extraction of spatial interrelations among the patches. In contrast, channel-mixing MLPs delve into the image’s channel-specific nuances, extracting pertinent features spanning various channels—such as the RGB components in an image.

After going through the mixer layers, Global Average Pooling is applied to the output to calculate the mean value of every spatial position within each channel. Finally, the output is classified to be either violent or non-violent after routing the resultant data through a layer that is fully connected, followed by activation of the Softmax function.

A salient advantage of the MLP-Mixer lies in its streamlined design. This design not only simplifies implementation but also strikes a harmonious balance between computational efficiency and performance, positioning it favorably against more complex, state-of-the-art models.

### 3.4. Sequential Image Collage

The dual-form dataset, denoted as *I*, comprises both image collages *S* and frames of the original videos *F* where S∩F=∅ and I=S∪F. Here, $S=\{S_{1},S_{2},\ldots ,S_{n}\}$ and $F=\{F_{1},F_{2},\ldots ,F_{m}\}$, with *m* and *n* representing the number of original video frames and image collages, respectively. The dimensions of an original video frame are 224×224. For the video frames to fit into a 3×3 image collage, nine consecutive frames of the original video will have to be resized into the dimensions of 224/3×224/3. Every image collage is then paired and accompanied by its nine original video frames as the input data as shown in Figure 2. *I* is then patch embedded before being fed into the proposed model. Patch embedded refers to the patch embedding concept that was proposed in the original paper of the vision Transformer [31] where it splits images into patches, flattens them, and projects them into dimensions of a fixed latent vector size
(1)X=PatchEmbedded(I),
where $X\in R^{S_{r}\times C}$. $R^{S_{r}\times C}$ denotes the resulting feature map, *C* is the designated dimension that is hidden, and $S_{r}$ is a series of image patches that do not overlap. $S_{r}$ is computed as $HW/P^{2}$ given that an input image has a patch size of P×P and resolution size of H×W.

Subsequent for patch embedding, the data then go through the operation of token mixing, followed by channel mixing in a mixer layer. Mathematically, this step unfolds as:(2)$$U_{\left[:,i\right]}=X_{\left[:,i\right]}+W_{2}\sigma \left(W_{1}L_{N}{\left(X\right)}_{\left[:,i\right]}\right)$$
and
(3)$$Y_{\left[j,:\right]}=U_{\left[j,:\right]}+W_{4}\sigma \left(W_{3}L_{N}{\left(U\right)}_{\left[j,:\right]}\right),$$
where the normalization of a layer is denoted as $L_{N}\left(\cdot \right)$, and the output of channel mixing and token mixing is denoted as *Y* and *U*. The channel and spatial positions are indicated as indices *j* and *i*.

Token mixing commences with normalizing feature map *X* using layer normalization, subsequently employing weight matrices $W_{1}$ and $W_{2}$ using linear transformations, and finally applying a function of non-linear activation σ such as ReLU [55]. *U* is then obtained by augmenting the processed output with the original input and later acts as the input of the channel mixing operation.

Conversely, the output from the token-mixing operation *U* is applied with layer normalization as the commencement of the channel-mixing operation. Weight matrices $W_{3}$ and $W_{4}$ are then applied to *U* by using linear transformations, followed by an activation function σ. The result is amalgamated with the initial output to produce *Y*. With the help of channel-mixing and token-mixing operations, the model is capable of capturing both local and global features from the violent video dataset.

## 4. Experimental Results

### 4.1. Datasets

A diverse array of datasets is trained in this paper to rigorously assess the efficacy of the SIC model, each tailored to distinct scenarios and behaviors.

The dataset of smart-city CCTV violence detection (SCVD): The SCVD dataset, as detailed in [56], serves as the primary benchmark. It encompasses three distinct classes: non-violent behavior, violent behavior involving weapons, and violent behavior not involving weapons. The dataset is characterized by varying clip counts across these categories, with 112 clips for weapon violence, 124 for violent behavior, and 248 for non-violent behavior. Notably, during the experimental phase, weapon violence and violent behavior are amalgamated into a singular category. This decision stemmed from the primary objective of discerning the mere presence or absence of violent acts within the imagery.The dataset of real-life violence situations (RLVS): This RLVS dataset, introduced in [57], offers a real-world glimpse into violent and non-violent scenarios. It comprises 1000 videos sourced from YouTube, evenly split into violent and non-violent categories. The violent segments predominantly feature actual altercations, such as street skirmishes, while the non-violent segments depict everyday activities—ranging from exercising and eating to walking.Hockey fight dataset: The hockey fight dataset, as documented in [17], is tailored specifically for violent behavior recognition within the context of professional ice hockey. It houses 1000 videos, meticulously curated from National Hockey League matches. These videos are evenly distributed, encompassing 500 clips each for violent and non-violent behaviors, providing a specialized lens into the nuances of aggression within a sports setting.

By leveraging these diverse datasets, the training regimen for the SIC model was designed to be comprehensive, ensuring robustness and adaptability across varied violent and non-violent contexts.

### 4.2. Hyperparameter Settings

The SIC network undergoes meticulous training to ensure optimal performance. The Adam optimizer, which is a widely recognized algorithm for deep learning, is employed with the setting of having the learning rate at $1\times 10^{-4}$ to facilitate stable convergence and efficient training. The MLP-Mixer, which is a state-of-the-art model renowned for its efficiency and performance in image processing tasks, is selected as the backbone architecture. To maintain consistency and harness the full potential of this architecture, the settings implemented in this paper are the same as the ones in the original paper B/16. A comprehensive overview of these settings is tabulated in Table 1 for reference. The image data fed into the network are standardized to a resolution of 224×224, a common choice for many vision tasks, ensuring a balance between computational efficiency and detailed feature extraction.

To further enhance the model’s resilience and generalization capabilities, the Mixup technique [58] is integrated as the data augmentation strategy. Mixup operates by blending pairs of images and their corresponding labels, synthesizing augmented training samples. This approach imbues the model with an enriched and diversified training set, effectively reducing overfitting and bolstering performance during the rigorous training regimen. By amalgamating these advanced techniques and strategies, this paper aims to cultivate a robust and adept SIC model primed for real-world applications.

### 4.3. Results

The efficacy and robustness of the SIC model were rigorously tested across three distinct datasets tailored to violent video recognition. These datasets include the hockey fight dataset [17], the real-life violence situations (RLVS) dataset [57], and the smart-city CCTV violence detection (SCVD) dataset [56]. Upon meticulous evaluation, detailed findings and metrics are encapsulated in Table 2. A key observation from these experimental outcomes is that the model excels in discerning and classifying violent events with remarkable precision.

A distinctive attribute of SIC lies in its adeptness at amalgamating both super images and original video frames. This fusion facilitates a holistic data representation, enabling the model to seamlessly integrate both spatial and temporal cues intrinsic to the video sequences. Furthermore, the synergy between super images and video frames amplifies the model’s depth of understanding, culminating in heightened accuracy and comprehensive insights into violent event recognition, thus underscoring its potential as a formidable tool in the domain of violent video analysis.

In addition to evaluating the effectiveness of SIC, a comprehensive comparison is conducted with several popular CNN and Transformer models to validate its performance. In Table 2, the results clearly demonstrate that the SIC method is capable of producing impressive outcomes across three diverse datasets. The proposed approach not only exhibits competitive accuracy but also performs exceptionally well in terms of the Rank-1 metric. This indicates that the SIC is capable of delivering reliable and high-quality results, making it a strong contender in the field.

Moreover, the efficiency of the SIC method is highlighted in Table 3, where its computational requirements are compared with other different approaches. The table shows that the proposed method outperforms others in terms of the floating point operations (FLOPs) and the number of parameters used. The reduced numbers of FLOPs and parameters signify that the SIC method is computationally efficient and resource friendly. This aspect is crucial as it allows the proposed model to achieve excellent performance while minimizing computational overhead.

Taken together, the outcomes showcased in both Table 2 and Table 3 provide a comprehensive evaluation of the SIC method. With fewer parameters and FLOPs compared to other state-of-the-art models, the proposed method is capable of achieving competitive accuracies. This underscores the method’s ability to perform efficiently with fewer resources while demonstrating its potential to compete with other advanced models. Not only does the proposed method operate effectively with less computational overhead, but it also delivers performance that is comparable to or even better than models that require more computational power. This highlights how effective and practical the SIC method is, making it a promising solution for various applications in the field of deep learning.

Furthermore, to demonstrate that the SIC architecture successfully learns violent behavior signatures, heatmaps are generated using the Grad-CAM (gradient-weighted class activation mapping) [63] visualization technique. These heatmaps give insights into how well the deep learning model SIC pays attention to different regions of the input image. In Figure 3, it is evident that the SIC model has accurately identified the parts of the image associated with physical contact, enabling a clear determination of the occurrence of violent behavior. This result is valuable because it helps us to understand the model’s decision-making process and provides valuable explanations for its predictions.

### 4.4. Ablation Study

#### 4.4.1. Different Combinations of Super Images

In the SIC method, the effect of three different combinations of super images (2×2, 3×3, and 4×4) is experimentally evaluated on the model performance. Table 4 demonstrates that the 3×3 combination outperforms the other two combinations in performance.

These results show that the super image combination has an important impact on the model’s performance. The 2×2 combination may not provide enough contextual information; hence, the performance of the model degrades. Conversely, 4×4 combinations may contain too much information, causing great difficulty for the model to comprehend effective feature representations. As the 3×3 combination resulted in having the best performance, this proves that the 3×3 combination is capable of providing the best effect in balancing contextual information and feature learning, in other words, giving the right amount of contextual information while avoiding excessive complexity.

#### 4.4.2. Temporal Shift of Super Image

In video analysis, maintaining the coherence of actions is very important for action recognition. Therefore, the method of temporally shifting the super image is explored to test its effect on the SIC method. The original super image is composed of consecutive video frames, while the newly shifted super image temporally overlaps the last six frames of the previous super image.

Table 5 is the comparison of the results using the original super image and shifted super image. Both methods used the same dataset for training and testing, and are evaluated using the accuracy and Rank-1 metrics to measure the model’s performance. The results of the experiment proved that using shifted super images for temporal displacement does not provide significant help to the SIC method, and does not increase the model’s ability to understand the coherence of actions.

#### 4.4.3. Different Patch Sizes of SIC

Patches are one of the key elements in the SIC method, which can capture local detail information in the image. However, different patch sizes may have different impacts on the performance and computational efficiency of the SIC method. Therefore, two sets of experiments are designed: 16×16 and 32×32 patch sizes. The same dataset is utilized for training and testing, and the difference between the two patch sizes is evaluated in terms of the model performance. In Table 6, the results showed that better results are obtained with a 16×16 patch. Thus, a 16×16 patch can provide more local detail information and help the model better capture subtle features. Moreover, the smaller size of each patch reduces the calculation operations to be performed on each patch, which can lead to a reduction in the amount of calculation.

## 5. Conclusions

A novel methodology for discerning violent behaviors within videos is introduced in this paper, leveraging the capabilities of the MLP-Mixer architecture in tandem with the approach of a sequential image collage (SIC) dual-form dataset. Distinctively, an efficient alternative to self-attention mechanisms and conventional convolutional operations is offered by the MLP-based framework. Its inherent computational simplicity renders it particularly apt for deployment in resource-constrained environments.

Central to our approach is the integration of both video frames and image collages (super image) within the SIC dual-form dataset. This comprehensive dataset amalgamation empowers the model to adeptly assimilate spatio-temporal intricacies inherent to violent actions, potentially augmenting its proficiency in violence recognition tasks.

Encouragingly, empirical evaluations showcased the SIC methodology’s competitive performance across three benchmark violence datasets, by achieving competitive accuracies while requiring fewer parameters and FLOPs compared to other state-of-the-art models. As a culmination of our efforts, we envisage that the insights and methodologies delineated herein will pave the way for enhanced real-world applications in the realm of violent video analysis.

## Figures and Tables

**Figure 1 sensors-24-01844-f001:**
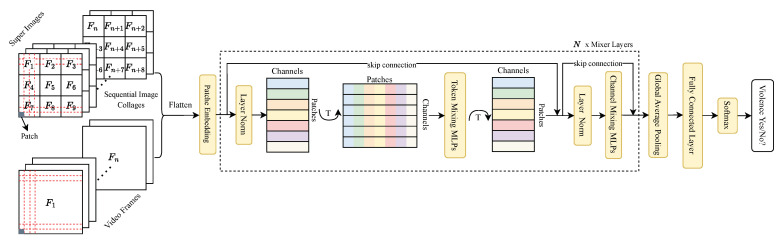
The proposed SIC model’s architecture. Each collage is composed of 3×3 video frames, where each video frame is denoted as $F_{n}$.

**Figure 2 sensors-24-01844-f002:**
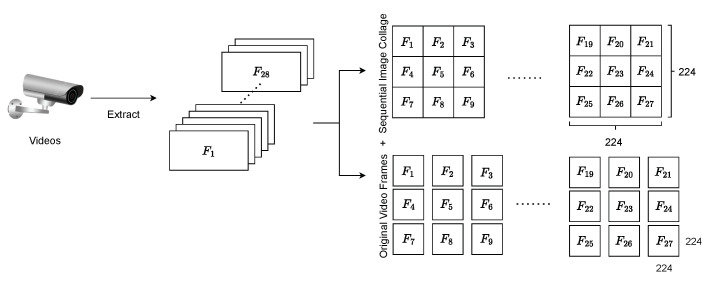
The process of grouping an image collage following a left-to-right and top-to-bottom order. (The image collage formed by aggregating several consecutive frames is denoted as $S_{n}$, while each frame of the original video is denoted as $F_{n}$).

**Figure 3 sensors-24-01844-f003:**
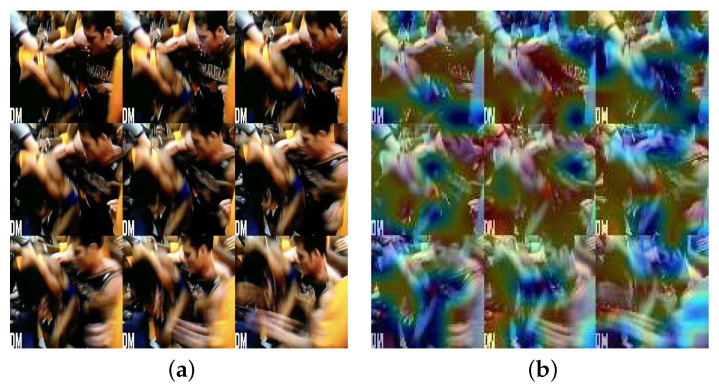
The GradCAM of the original violent picture output from the SIC. (**a**) Violence picture; (**b**) GradCAM picture.

**Table 1 sensors-24-01844-t001:** MLP-Mixer settings.

Specification	S/16
Patch resolution P×P	16 × 16
Number of layers	8
Sequence length *S*	196
Hidden size *C*	512
MLP dimension $D_{C}$	2048
MLP dimension $D_{S}$	256
Parameters	18 M

**Table 2 sensors-24-01844-t002:** Performance comparison of models evaluated on different datasets.

Type	Method	Hockey Fight		RLVS		SCVD	
		Acc.	R-1	Acc.	R-1	Acc.	R-1
Transformer-based	ViT/B16 [31]	97.4	97.4	96.1	96.0	90.4	89.2
	Swin-B [52]	97.8	97.8	95.5	95.5	90.4	89.4
	PVTv2 [59]	97.3	97.3	97.1	97.1	90.6	89.5
CNN-based	ResNet50 [60]	97.1	97.1	96.9	96.9	90.1	89.0
	ResNet101 [60]	98.2	98.2	97.1	97.0	91.0	90.0
	VGG16 [61]	97.2	97.2	96.0	95.9	89.2	88.2
	InceptionV4 [62]	97.3	97.3	97.1	97.1	90.1	89.1
MLP-Mixer S/16	MLP-Mixer [34]	96.0	96.0	88.4	88.4	74.0	74.0
MLP-Mixer B/16	MLP-Mixer [34]	96.9	96.9	89.6	89.6	76.5	76.5
MLP-based S/16	SIC (Ours)	97.5	97.5	96	95.9	90.7	89.7

**Table 3 sensors-24-01844-t003:** Comparison of parameters and FLOPs.

Method	Scale	Params	FLOPs
ViT/B16 [31] (ICLR 2021)	224 × 224	85.6 M	16.86 G
Swin-B [52] (ICCV 2021)	224 × 224	86.6 M	15.16 G
Resnet50 [60] (CVPR 2016)	224 × 224	23.5 M	4.13 G
Resnet101 [60] (CVPR 2016)	224 × 224	42.5 M	7.86 G
PVTv2 [59] (Comput. Vis. Media 2022)	224 × 224	24.8 M	3.89 G
VGG-16 [61] (ICLR 2015)	224 × 224	134.26 M	15.46 G
InceptionV4 [62] (AAAI 2017)	224 × 224	41.14 M	6.15 G
MLP-Mixer S/16 [34] (NeurIPS 2021)	224 × 224	18 M	3.78 G
MLP-Mixer B/16 [34] (NeurIPS 2021)	224 × 224	59.88 M	12.61 G
SIC (ours)	224 × 224	18 M	3.78 G

**Table 4 sensors-24-01844-t004:** The results of the SIC using different combinations of super image on three datasets.

Combination	Hockey Fight		RLVS		SCVD	
	Acc.	R-1	Acc.	R-1	Acc.	R-1
2×2	97.0	97.0	95.4	95.3	89.3	88.2
3×3	97.5	97.5	96.0	95.9	90.7	89.7
4×4	97.0	97.0	94.2	94.1	89.5	88.5

**Table 5 sensors-24-01844-t005:** The results of the influence of two different super images on the SIC method.

	Hockey Fight		RLVS		SCVD	
	Acc.	R-1	Acc.	R-1	Acc.	R-1
super image	97.5	97.5	96.0	95.9	90.7	89.7
shift super image	97.1	97.1	95.7	95.7	89.3	88.2

**Table 6 sensors-24-01844-t006:** The results of SIC using different patch sizes on three datasets.

Patch Size	Hockey Fight		RLVS		SCVD	
	Acc.	R-1	Acc.	R-1	Acc.	R-1
16×16	97.5	97.5	96.0	95.9	90.7	89.7
32×32	96.9	96.9	95.6	95.6	90.3	89.2

## Data Availability

The original data presented in the study are openly available in SCVD dataset at https://www.kaggle.com/datasets/75806dc0d1bc0fccd0cedaf117979ffa2f2ae5c3c7af3cdd78b9f4cc14d96013, RLVS dataset at https://www.kaggle.com/datasets/mohamedmustafa/real-life-violence-situations-dataset/ and Hockey fight dataset at https://github.com/tintybot/CNN-BiLSTM-Model.

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
