# Peer review of "Violent Video Recognition by Using Sequential Image Collage"

_sensors, 2024, doi:10.3390/s24061844_

Round 1

Reviewer 1 Report

Comments and Suggestions for Authors

The article proposes an MLP mixer architecture called sequential image collage for identifying violent activities in videos.

Their last sentence in the abstract about superiority of their model contradicts with the results in Table 3.

   + Hockey database. best results recorded by transformer-based Swin-B [52]

   + RLVS database: Many transformer-based and CNN based outperformed the proposed model[31,59,60,62].

   + SCVD database: best results recorded by CNN-based ResNet101 [60].

Could you please discuss these findings and what are the benefits of the proposed model over the other approaches.

- Are the settings in Table 1 the optimal ones? for example, did your try different patch sizes or # of layers ?

- Table 2 is redundant. All results in Table 2 are already presented in Table 3. 

- Did you investigate using different image collage sizes other than 3x3?

- Reference section needs revision for inconsistency in formatting styles.

The article is a good study regarding violent video recognition problem. I believe with the above minor corrections and suggestions, the article will be qualified to be published in this journal.

Reviewer 2 Report

Comments and Suggestions for Authors

In this research, a special type of dataset to be fed into the MLP-mixer called sequential image collage (SIC) is proposed.

in the  conclusion, you mention that "Central to this approach is the integration of both video frames and image collages (super image) within the SIC dual-form dataset."

However, the outperformance of this integration is not clearly presented in this paper.

contributions of your paper compared to papers in section 2 are not clear to me.

table 3 does not show the outperformance of your method clearly.

computational load of your method needs to be compared with the state of the arts. is it possible to run your method in real time?

Comments on the Quality of English Language

ok

Reviewer 3 Report

Comments and Suggestions for Authors

The research in field of violent video recognition looks relevant and interesting nowadays. The authors propose to use temporal context information as input of the known MLP-Mixer architecture. 

The remarks could be formulated as follows.

1. Not all of the text matches the journal «Sensors» template. For example, the paragraphs after lines 231 and 234 don’t have the numeration of lines, and the numeration of the Reference Section doesn’t exist. 

2. The authors should provide a description of the operation “PatchEmbedded”.

3. The number of patches in the proposed dual-form dataset (Subsection 3.4) needs to be described in detail. It is clear that the dataset includes a set of video frames and a set of image collages (first paragraph in Subsection 3.4), but in the text (lines 233-234), the Sr is computed as the number of patches in a single image.

4. The authors should compare the performance of the proposed method with the original MLP-Mixer [Tolstikhin, I.O.; Houlsby, N.; Kolesnikov, A.; Beyer, L.; Zhai, X.; Unterthiner, T.; Yung, J.; Steiner, A.; Keysers, D.; Uszkoreit, J.; Lucic, M.; Dosovitskiy, A. MLP-Mixer: An all-MLP Architecture for Vision. Advances in Neural Information Processing Systems; Ranzato, M.; Beygelzimer, A.; Dauphin, Y.; Liang, P.; Vaughan, J.W., Eds. Curran Associates, Inc., 2021, Vol. 34, pp. 24261–24272]. 

5. Since the quality of the proposed method differs not much from the others mentioned, the benefit of the proposed method is fewer parameters, but the number of parameters in the original MLP-Mixer is the same. So, the advantage of the proposed method is not clear.

Round 2

Reviewer 2 Report

Comments and Suggestions for Authors

I am satisfied with revision.

Comments on the Quality of English Language

ok

Reviewer 3 Report

Comments and Suggestions for Authors

The research in field of violent video recognition looks relevant and interesting nowadays. The authors propose to use temporal context information as input of the known MLP-Mixer architecture.

The comments 1 and 2 are addressed adequately.

Concerning the comment 3. The authors propose to feed both image collage (noted as S) and video frames (noted as F) as described in lines 226-229. However, the parameter Sr refers to the number of patches obtained from single image. So, the technique of the image collage and video frames joining (noted as I) is not clear in the proposed paper. Since only that aspect looks like a novel in the proposed paper, the authors should add more detailed description of this preprocess algorithm and estimate its computational complexity.

Concerning the comments 4 and 5. The added parameters in Table 3 look like the parameters for original MLP-Mixer with specification B/16, but the MLP-Mixer with specification S/16 is used in the proposed paper. So, including the comparison of original MLP-Mixer B/16 and used MLP-Mixer S/16 isn’t correct in Table 3. The comparison accuracy between proposed MLP-Mixer S/16 with SIC technology and original MLP-Mixer S/16 without proposed SIC technology would look more appropriate. So, the advantage of the proposed method is not clear. It’s advised to compare two identical MLP-Mixers differing only in usage of the proposed SIC technology.
